# Gait stability in ambulant children with cerebral palsy during dual tasks

**Sophie Wist**[1,2]*, **Lena Carcreff**[2], **Sjoerd M. Bruijn**[3], **Gilles Allali**[4,5], **Christopher J. Newman**[6], **Joel Fluss**[7], **Stéphane Armand**[2]

**1** Zürcher Hochschule für Angewandte Wissenschaften, ZHAW, Institut für Physiotherapie, Winterthur, Switzerland, **2** Kinesiology Laboratory, Geneva University Hospitals and University of Geneva, Geneva, Switzerland, **3** Department of Human Movement Sciences, Vrije University Amsterdam, Amsterdam, Netherlands, **4** Department of Neurology, Geneva University Hospitals and University of Geneva, Geneva, Switzerland, **5** Leenaards Memory Center, Lausanne University Hospital and University of Lausanne, Lausanne, Switzerland, **6** Pediatric Neurology and Neurorehabilitation Unit, Department of Pediatrics, Lausanne University Hospital and University of Lausanne, Lausanne, Switzerland, **7** Pediatric Neurology Unit, Children's Hospital, Geneva University Hospitals, Geneva, Switzerland

* sophie.wist@gmail.com

## Abstract

### Aim

The aim of this cross-sectional study was to measure the effect of dual tasks on gait stability in ambulant children with cerebral palsy (CP) compared to typically developing (TD) children.

### Methods

The children of the CP (n = 20) and TD groups (n = 20) walked first without a dual task, then while counting forward and finally while alternatively naming fruits and animals ($DT_{f/a}$). They then completed the same cognitive exercises while sitting comfortably. We calculated the distance between the foot placement estimator (FPE) and the real foot placement in the anterior direction ($D_{FPE}AP$) and in the mediolateral direction ($D_{FPE}ML$) as a measure of gait stability, in a gait laboratory using an optoelectronic system. Cognitive scores were computed. Comparisons within and between groups were analysed with linear mixed models.

### Results

The dual task had a significant effect on the CP group in $D_{FPE}AP$ and $D_{FPE}ML$. The CP group was more affected than the TD group during dual task in the $D_{FPE}ML$. Children in both groups showed significant changes in gait stability during dual tasks.

### Interpretation

The impact of dual task on gait stability is possibly due to the sharing of attention between gait and the cognitive task. All children favoured a 'posture second' strategy during the dual task of alternatively naming animals and fruits. Children with CP increased their mediolateral stability during dual task.

**Data Availability Statement:** All relevant data are within the paper and its Supporting Information files.

**Funding:** This work was supported by La Fondation Paralysie Cerebrale (Paris, France, https://www.fondationparalysiecerebrale.org/), no grant number is available. Sjoerd M.Bruijn was funded by a VIDI grant (016.Vidi.178.014) from the Dutch Organization for Scientific Research (NWO, https://www.nwo.nl/en). The funders had no role in study design, data collection and analysis, decision to publish, or preparation of the manuscript.

**Competing interests:** The authors have declared that no competing interests exist.

## Introduction

Cerebral palsy (CP) is a permanent neurological disorder caused by non-progressive brain lesions occurring before, during or in the months after birth [1]. With a prevalence of 1.77 per 1000 live births in Europe, CP is the most common cause of significant motor impairment in children [2]. Stability is reduced in children with CP because of motor and cognitive impairments, which can lead to falls [3]. Most falls happen while walking, one of the most frequent motor activities [4]. If a child is able to ambulate independently, she/he will consequently increase her/his risk of falling [3]. Children with disabilities are more exposed to concussions when falling than non-disabled children, who generally suffer from less severe damage, such as upper limb injuries [5]. Petridou et al. [5] also found that children with disabilities experience more falls at school or at home during necessary activities in comparison to accident rates in children with typical development (TD), which happen mostly during leisure activities.

Gait stability measurements assess the ability to walk without falling and are dependent on a person's neuromuscular ability not to fall when exposed to sources of disturbance [4]. Several measurements or estimators of gait stability have been proposed, such as the foot placement estimator (FPE) [6], variability measures [7], extrapolated centre of mass [8], maximum Lyapunov exponent [9] and maximum Floquet multiplier [10].

Among these parameters, the FPE, which estimates where the foot should be placed to come to a standstill state at mid-stance [4], seems relevant for children with CP. This measure was developed by Wight in 2008, firstly for robotic applications [11]. It was then tested on healthy humans under different walking speeds and activities [6,11–13]. It has also been used for children with CP [14,15]. It differentiates them from children with TD [15]. For example, Bruijn et al [15] showed, that children tend to place their feet near the FPE in the anterior position while walking at self-selected speed, whereas when the walking speed increases, they tend to increase the distance between the real foot placement and the FPE. The FPE method is the only gait stability measure to integrate the assumption of conservation of angular momentum, integrating in its calculation the loss of energy and velocity which is present in human gait [16]. Moreover, only a small number of strides are needed to get valid results, whereas numerous strides are needed with other methods such as Floquet multipliers or variability measures [15].

A combination of two activities carried out at the same time, for example talking while walking, is referred to as a dual-task, and is one of the sources of disturbance that occurs in daily-life [17]. The central capacity sharing model describes that when two tasks are processed at the same time, both tasks will be affected and potentially worsened [18]. It implies that gait requires attention, and that it is not an automatic process [18]. Most interferences at the cortical level appear in single-limb stance, when postural adjustments are planned [19]. Boonyong et al. [20] found a reduced anteroposterior centre of mass (CoM) sway in children with TD while walking under dual-task, and suggested that they modified their walking speed and step length to improve stability. In children with CP, decreased gait speed, stride length [21,22] and anteroposterior trunk acceleration, as well as increased lateral trunk acceleration, have been shown during dual tasks compared to unchallenging gait [17,21,22]. To our knowledge, there is no previous study measuring gait stability using the FPE during dual tasks in children with CP.

This study aimed to investigate gait stability during dual tasks in children with CP with the use of the FPE. We firstly hypothesised that children with CP would show a larger distance between real foot placement and FPE under dual tasks than under simple task, in order to stabilise their gait while performing a concurrent cognitive task. Our second hypothesis was that the dual-task effect would be higher in children with CP than in children with TD. These

findings could lead to a better understanding of gait stability in children with CP in ecological situations.

## Method

The design was a cross-sectional study. It was conducted in a clinical setting. The study was approved by the ethical committee of canton Geneva in 2015 (CCER-15-203).

### Participants

The study population consisted of children with CP between the ages of 8 and 16 years with level I or II according to the Gross Motor Function Classification System (GMFCS) [23], as well as age and sex-matched children with TD.

The sample consisted of two groups, children with CP (CP group) and age and sex-matched children with TD (TD group). Children were age matched with peers, with a range of ±1.5 years. This corresponds to the approximate age range present in a classroom with children who have the same level of semantic fluency [24].

In order to be included in the CP group, children had to be able to walk a minimum of 50 metres without any assistance and had to follow a regular school curriculum. Exclusion criteria were an intelligence quotient (IQ) below 80 and behavioural problems. For the TD group, the exclusion criteria were an IQ lower than 80, behavioural problems, and any other issues affecting gait or cognitive performance.

Children with CP were recruited among patients followed at Geneva University Hospitals (HUG) and patients sent to the laboratory for gait analysis. Children with TD were recruited through investigators' and patients' families and friends. Every child, as well as their parents, read and signed an informed consent.

Sample size calculation was based on the effect size of a study measuring gait speed during simple and dual-task in a similar population [21]. They found an effect size of 0.97 on the dual-task constraint of identification of a common sound. With an α error probability of 0.05 and a β = of 0.80, a sample size of 14 children per group was required (computed by G*power). Because of the high heterogeneity in age and the different CP types, we decided to increase this number to 20 children per group.

### Protocol

The data collection was performed in the Kinesiology Laboratory at HUG between February 2016 and March 2019. The trajectory of 35 reflective markers positioned according to the Conventional Gait Model [25] was registered while walking, by a 12-camera optoelectronic system (Oqus 7+, Qualisys, Gothenburg, Sweden). The examination started with the height, weight and lower limb strength. Then they walked 10 metres barefoot at a self-selected pace during 3 trials. The first trial was the simple motor task. During the next two trials, they had to perform cognitive tasks in a random order while walking, with a 30-second break in between. The easiest task consisted in counting out loud forward from zero ($DT_{count}$). The fluency task, which was considered as the hardest cognitive task, was alternatively listing fruits and animals ($DT_{f/a}$). The time taken to execute the dual-task trials was recorded. This was followed by the measurements of cognitive performance while the participant was sat comfortably on a chair with back-and armrests. The simple cognitive tasks were the same as under dual-task constraint. The patients were granted as much time for these tasks as they used to execute the walking dual tasks.

## Outcomes

The primary outcome was the FPE [6] which was separated into 2 parts: the distance from the foot to the FPE in the anteroposterior direction ($D_{FPE}AP$) and in the mediolateral direction ($D_{FPE}ML$). Firstly, we computed the centre of mass (CoM) from 14 segments weighted average CoM [26]. Those estimations relied on an anthropometric model [27,28] that uses the participants' mass and height. The inertia of each segment was calculated using the total body CoM as a reference [6]. The ground projection of the CoM was calculated as $CoM_p$ [26]. From the $CoM_p$, the angular momentum of the total body and the plane of progression were determined [13]. The FPE, which represents the ideal placement of the foot to guarantee stability for an inverted pendulum [15], was computed using the inverted pendulum model and the total body inertia. The distance between FPE and the most anterior marker placed on the 2nd metatarsal ($D_{FPE}AP$) and the distance between FPE and the most lateral point of the foot, which was either the lateral malleolus marker or the 5[th] metatarsal marker ($D_{FPE}ML$) were used for the statistical analysis [4,13,29]. Positive values of $D_{FPE}AP$ and $D_{FPE}ML$ indicate that the foot is placed respectively behind and medial to the calculated FPE, as illustrated on Fig 1. In the case of positive values, the CoM movement cannot be stopped within a step; the more negative the values are, the more likely this movement can be stopped, and the more stable the subject [15]. We used the results of the affected leg in case of unilateral CP and chose the most affected leg for children with diplegia, based on muscular strength of the lower limb during clinical examination. This was tested using the manual muscle testing (MMT). In the TD group all FPE results were arbitrarily taken from the right leg.

Secondary outcomes were cognitive scores and gait parameters (speed, speed normalized by leg length, cadence, step length and step width computed from the marker trajectories). Correct answers of the cognitive tasks were counted per second for a cognitive score. Invented

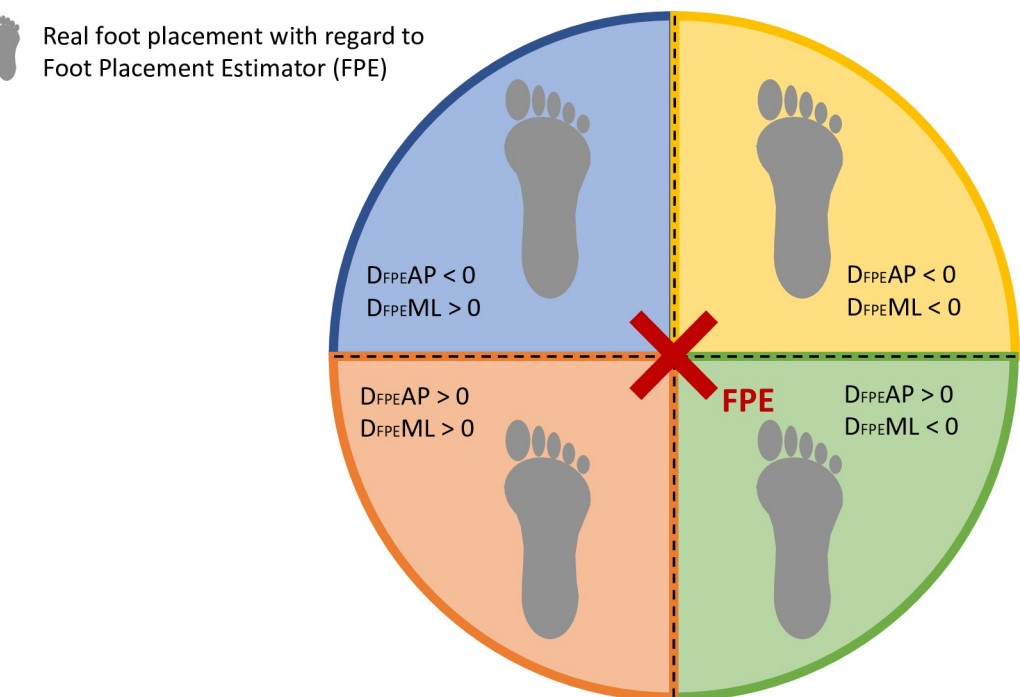

**Fig 1. Foot placement estimator values.** Values of the distance between the real foot and the foot placement estimator ($D_{FPE}$) in the anteroposterior (AP) and mediolateral directions (ML) with regards to 4 areas of real foot placement.

words and repetitions were excluded, as well as omissions of numbers while counting. The FPE and the gait parameters were computed using MATLAB (MATLAB 2016b, MathWorks, MA, USA) for each trial [30].

## Statistical analysis

The statistical analyses were executed using R v.4.0.4 and the RStudio interface (v.1.4.1, Rstudio Team). T tests were performed to assess the homogeneity of the groups regarding anthropometric characteristics. To assess how the $D_{FPE}$ varied between Groups and Task individual linear mixed models were fit for the $D_{FPE}$ values. This was calculated for each plane (AP and ML) and regressed on Group and Task. When the interaction was found significant the model with interaction of the Group by Task was used. The significance of the interaction was assessed by an ANOVA between the models with (Eq 1) and without (Eq 2) interaction. Additionally, random intercepts were fit for each pair of individuals (Pair ID) to account for pair matching. To determine whether the interaction effect remained significant when covariates were included, normalized gait speed was added to each model and retained if significant (p<0.05). The normalized gait speed was centered about its mean value across all individuals and Tasks.

$$\text{Model with interaction}: \ D_{FPE} \sim \text{Group} * \text{Task} + (\text{speed} - \text{mean speed})|\text{Pair ID} \quad \text{(Eq 1)}$$

$$\text{Model without interaction}: \ D_{FPE} \sim \ \text{Group} + \text{Task} + (\text{speed} - \text{mean speed})|\text{Pair ID} \ \text{(Eq 2)}$$

The Task effect was used to verify our first hypothesis which was that the CP group would demonstrate a larger $D_{FPE}$ under DT as compared to simple task. The Group by Task effect was used to examine our second hypothesis stating that the CP group would demonstrate significantly greater changes in $D_{FPE}$ from the simple task to DT as compared to the TD group. Regression coefficients, confidence intervals and p values were calculated for each effect. The TD group and simple task were the reference effects in each model.

There was no missing data. The mean age, weight and height of the participants of both groups were reported. Mean values of $D_{FPE}AP$, $D_{FPE}ML$, walking speed, cadence, step length, step width and cognitive scores were reported per group under the three Tasks, including the simple walking task and the two dual-task trials. The mean values of the cadence, step length and step width were reported to get a better understanding of the $D_{FPE}$ and its values but were not statistically analysed. The data distribution of $D_{FPE}AP$ and $D_{FPE}ML$ in each group and under each Task was visually controlled.

For the cognitive scores, the data distribution in each group and under each Task was analysed using skewness and kurtosis z-scores [31,32]. When skewness and kurtosis z-scores were beyond 1.96, the data was qualified as not normally distributed [31] and were transformed with a logarithm (log10). When the distribution was normal the data were compared using a T-test. When the data was not normal even after the logarithmic transformation, a non-parametrical paired test (Wilcoxon) was used. The level of significance was set at 0.05 for all analyses.

The methodology was controlled using the STROBE checklist for observational studies.

## Results

A total of 40 children responding to the selection criteria were included in this study. Participants were aged between 8 and 16 years old at the measurement time in the CP group and between 9 and 16 years in the TD group (Table 1). The CP group included children with spastic unilateral (n = 13, the affected side was left for 6 of them and 7 right) and bilateral (n = 7,

**Table 1. Mean, standard deviation and range for the general characteristics of the participants.**

|  | CP (n = 20) | | | TD (n = 20) | | |
|---|---|---|---|---|---|---|
|  | **Mean** | **SD** | **Range** | **Mean** | **SD** | **Range** |
| **Age (y; m)** | 12y 6m | 2y 4m | 8y – 16y 9m | 12y 6m | 2y 4m | 9y 1m–16y 8m |
| **Height (m)** | 1.55 | 1.57 | 1.24–1.90 | 1.52 | 1.42 | 1.27–1.77 |
| **Weight (Kg)** | 51.35 | 22.6 | 21–107 | 41.35 | 12.14 | 24–65 |

Abbreviations: SD: Standard deviation; y: Years; m: Months; CP: Cerebral palsy; TD: Typical development.

the more affected limb was left for 4 and right for 3) CP, 17 of them had a GMFCS level of I and 3 had a GMFCS level of II. None of the children presented dysarthria. The two groups had similar weight (p = 0.09) and height (p = 0.53). Further details are available in Table 1.

On average, $D_{FPE}AP$ and $D_{FPE}ML$ were negative in both groups, meaning that the children placed their feet further and more lateral than the FPE respectively in the anteroposterior and the mediolateral directions. Fig 2A and 2B show the distribution of the two primary outcomes $D_{FPE}AP$ and $D_{FPE}ML$, in each task and each group.

The ANOVA outputs showed that the global interaction between Groups and Tasks was not significant for $D_{FPE}AP$ (p = 0.434), meaning that the dual tasks had similar effects on both groups. Table 2 reports the differences between groups, supposedly the same for each task by the model without interaction (Eq 1), and between Tasks, supposedly the same for each group. $D_{FPE}AP$ was significantly lower during the dual tasks, than during the simple task for both groups (p = 0.012 in DT Count, and p<0.001 in DT Animals), revealing a significant decrease of $D_{FPE}AP$ with the difficulty of the task (Fig 2A).

The interaction was statistically significant for $D_{FPE}ML$ (p = 0.008), meaning that the dual task effect was not equal between the groups. The model (Eq 2) with interaction was thus performed. Table 3 reports the mean differences between Groups and Tasks. The main difference with $D_{FPE}AP$ is that there was no significant difference between Groups during the simple task.

Normalized gait speed was found to significantly contribute to $D_{FPE}$ in the ML direction and, more importantly, in the AP direction (Table 2). Indeed, for an increase of $0.1s^{-1}$ the $D_{FPE}AP$ increases of 18.7 mm and the $D_{FPE}ML$ increases of 1.7 mm.

## Gait parameters

The mean speed and step length were lower in children with CP than in children with TD. Meanwhile, the mean step width was higher in the CP group than in the TD group. The mean speed and cadence lowered and the steps shortened in both groups with the dual task difficulty. The step width did not change during all tasks in the TD group while it became wider in the CP group when the difficulty of the cognitive task increased. The details are presented in Table 3.

## Cognitive scores

In both groups, during $DT_{count}$, children gave significantly more answers during the simple cognitive task (sitting) than during dual-task (CP: 0.099 (0.135) log10(answer/s), $p = 0.004$; TD: 0.1 (0.126) log10(answer/s), $p = 0.002$). It was not the case with $DT_{f/a}$, in which differences were not significant. Also, the TD group gave more answers per second than the CP group in every task. This difference was significant only during the fluency simple task (-0.107 (0.202) answer/s., p = 0.029).

D<sub>FPE</sub>AP

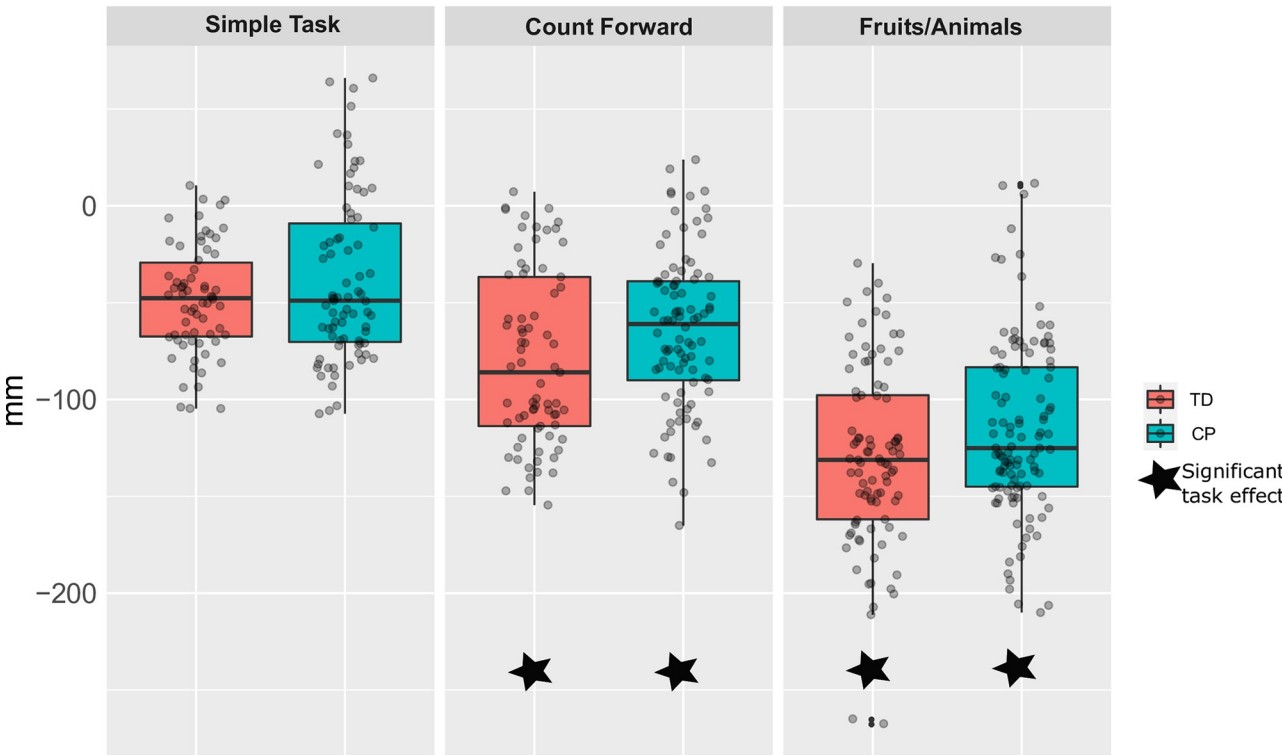

D<sub>FPE</sub>ML

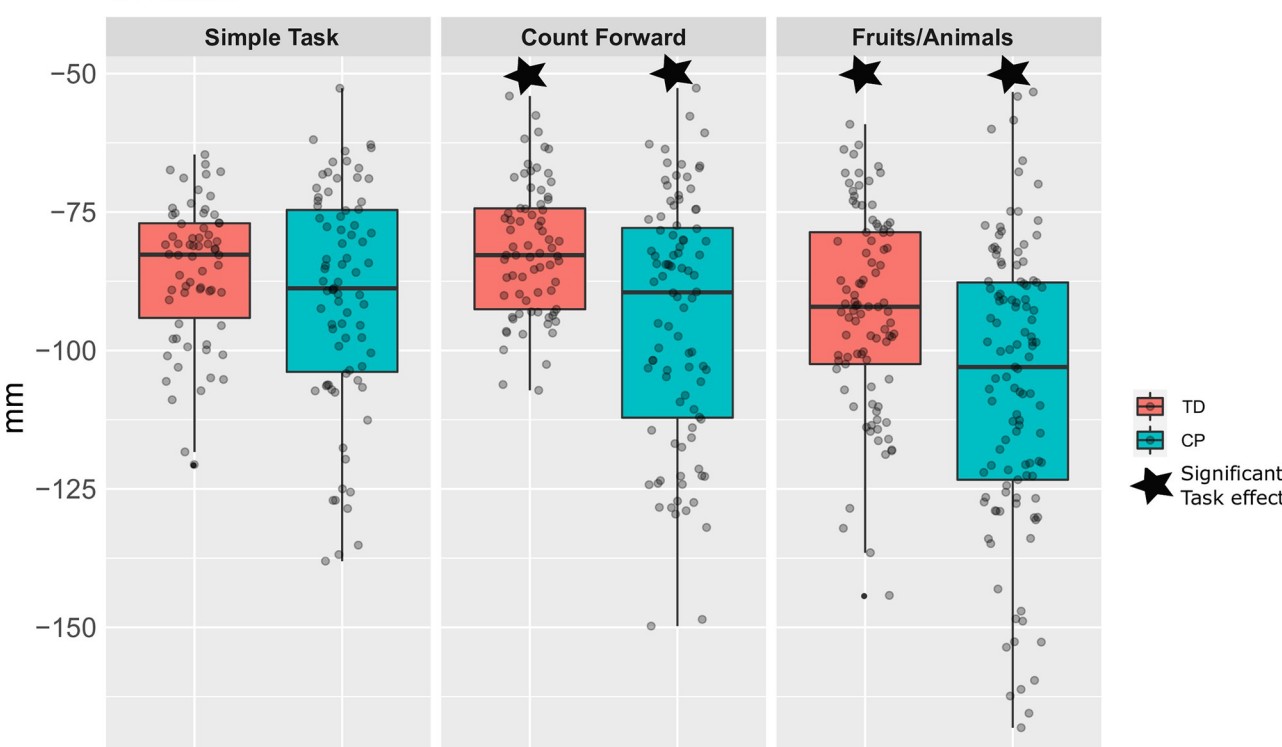

**Fig 2.** A, B. Gait stability in the anteroposterior ($D_{FPE}AP$) and mediolateral ($D_{FPE}ML$) directions. To allow a better visualisation of the results, we used the interquartile range (IQR) and mediane represented per task and per group, with each grey spot representing a participant. Black star represents a significant difference between the simple and the dual task. Red star represents a significant difference between the cerebral plasy (CP) and typical development (TD) groups.

## Discussion

We examined gait stability in children with CP during dual tasks. The main finding was that the CP group walked with a more stable gait under dual-task constraint than under simple gait task in both anteroposterior and mediolateral directions. In comparison, in the TD group, a task effect was observed in anteroposterior direction for both tasks, but not in the mediolateral direction. Our first hypothesis, which expected that children with CP would show a longer distance between real foot placement and FPE (a more stable gait) under dual tasks was verified.

In addition, a significant group effect was observed in $D_{FPE}ML$ for the counting task and the fluency task. Therefore, we could validate our second hypothesis, which stipulated that the impact of dual task would be higher in children with CP than in children with TD. Overall, these results show that the children with CP and with TD tend to stabilise their gait when under dual tasks. However, children with TD had to significantly modify their gait only in the anteroposterior direction, which is highly influenced by gait speed. Children with CP had to modify their stability in both directions. Finally, we showed that FPE can detect small changes in the gait of populations of children with CP and with TD. We expected certain differences in gait parameters between both groups. For all three tasks the step length was consistently lower in the CP group. Moreover, the difference between both groups became greater under increased dual-task difficulty. However, we observed that both groups had similar cadence, that decreased similarly in each group when the task became more difficult. It is therefore possible that in children with CP the spatial components of movement programming are more strongly impacted by dual-tasks than its temporal components, when compared to TD children.

Our results are in agreement with those of other studies. For example, Carcreff et al. [22] observed a similar dual-task cost between groups, on walking speed, stride length, hip range of motion, stride time and heel clearance. However, they obtained significant between-groups differences in the most difficult tasks' cost on the walk ratio (ratio step length/cadence) [22,33]. In our study, during both dual tasks, the CP group had a significantly lower $D_{FPE}ML$ than the TD group which was not significantly affected. This means that the CP group

**Table 2. Results of linear models for the $D_{FPE}$ (AP and ML).**

| | $D_{FPE}AP$ | | | $D_{FPE}ML$ | | |
|---|---|---|---|---|---|---|
| | Regression coefficient [CI] | | p value | Regression coefficient [CI] | | p value |
| Intercept (TD—Simple task) | -92.1 | **[-102.5;-81.7]** | **<0.001** | -90 | **[-96.8;-83.2]** | **<0.001** |
| Group | 31.8 | **[26.8;36.8]** | **<0.001** | -2.0 | [-7.1;-3.1] | 0.441 |
| Task (Count) | -8.0 | **[14.2;1.8]** | **0.012** | 5.8 | **[0.6;10.9]** | **0.029** |
| Task (Animals) | -19.4 | **[-27.9;-10.8]** | **<0.001** | -0.7 | [-6.7;5.4] | 0.830 |
| Group x Task (Count) | | | | -7.7 | **[-14.4;-1.0]** | **0.025** |
| Group x Task (Animals) | | | | -9.8 | **[-16.2;-3.5]** | **0.003** |
| Group x Task for CP group (Count) | | | | -1.9 | [-6.7;2.8] | 0.420 |
| Group x Task for CP group (Animals) | | | | -10.5 | **[-16.2;-4.8]** | **<0.001** |
| Normalized speed | 186.6 | **[165.7;207.4]** | **<0.001** | 16.8 | **[4.6;29.0]** | **0.008** |

TD: Typically developing; CP: Cerebral palsy; CI: Confidence interval.

**Table 3. $D_{FPE}$ and gait parameters during simple and dual tasks in children with CP and TD.**

| | | TD | CP |
|---|---|---|---|
| **$D_{FPE}AP$ (mm)** | **Simple gait** | -48.6 (28.0) | -38.2 (42.9) |
| | **$DT_{count}$** | -79.3 (45.3) | -115.2 (46.1) |
| | **$DT_{f/a}$** | -129.9 (46.9) | -115.2 (46.1) |
| **$D_{FPE}ML$ (mm)** | **Simple gait** | -85.8 (12.6) | -90.4 (19.9) |
| | **$DT_{count}$** | -82.0 (11.8) | -94.5 (22.1) |
| | **$DT_{f/a}$** | -92.7 (17.7) | -106.1 (25.6) |
| **Normalized speed ($s^{-1}$)** | **Simple gait** | 0.79 (0.09) | 0.68 (0.13) |
| | **$DT_{count}$** | 0.68 (0.11) | 0.57 (0.14) |
| | **$DT_{f/a}$** | 0.47 (0.14) | 0.37 (0.16) |
| **Cadence (step/min)** | **Simple gait** | 116.04 (7.95) | 116.32 (13.76) |
| | **$DT_{count}$** | 105.13 (13.98) | 108.13 (15.16) |
| | **$DT_{f/a}$** | 82.95 (19.88) | 82.84 (19.67 |
| **Step length (m)** | **Simple gait** | 0.62 (0.06) | 0.55 (0.08) |
| | **$DT_{count}$** | 0.59 (0.06) | 0.51 (0.1) |
| | **$DT_{f/a}$** | 0.5 (0.07) | 0.41 (0.08) |
| **Step width (m)** | **Simple gait** | 0.07 (0.02) | 0.09 (0.03) |
| | **$DT_{count}$** | 0.07 (0.02) | 0.1 (0.04) |
| | **$DT_{f/a}$** | 0.08 (0.03) | 0.13 (0.04) |

Mean (standard deviation) are presented for the simple task, the counting dual task ($DT_{count}$) and the fluency dual task ($DT_{f/a}$). The speed is normalized by leg length.second; min: Minute; m: Meter.

increased their stability while the TD group preserved its normal gait pattern. Tracy et al. [34] also found a "conservative stability strategy" in children with CP during different dual tasks activities, CP group had more lateral stability than children with TD. Finally, Katz Leurer et al. [21] found that children with CP tend to be affected by smaller changes than TD, because of lower basic motor capabilities and reduced attention, which matches our findings.

The fact that both gait parameters and cognitive scores were affected under dual-task constraint in the $DT_{count}$ in both groups could be explained by the central capacity sharing model [35]. This model explains how the central nervous system shares its capacity between both tasks and therefore both are impaired. For the fluency task, we witnessed nearly no cognitive score change between $DT_{f/a}$ and the fruit and animal listing simple task. It seems that children in both groups adopted an adapted version of 'posture second' strategy [36], which could be described as "mobility second". Due to the difficulty of the cognitive task which required more attention and concentration, they kept their cognitive scores at the same level but had to secure their stability in both directions. In our study, children showed their ability to adapt their gait towards a more stable pattern. These results are similar to the ones of Reilly et al. [37], who tested stability during dual tasks while standing in children with CP. They found that their postural control was impacted by the dual task [37]. In children with ataxia, the postural control was more impacted by the supplementary task and their results in the cognitive tasked were lower in comparison with children with spastic CP [37]. Those results, put together with ours, raise the question of whether the difficulty of the cognitive task is the cause, the baseline level of stability or the type of underlying brain lesion that most impact the dual tasks capacities of children with CP. The meta-analysis of Roostaiei et al. [38], described that the response of children with CP to dual task is highly influenced by their neuromuscular impairments.

In this study, we showed that walking speed and gait stability are linked, mostly in the anteroposterior direction. Carcreff et al. [22] showed that the CP group reduced their cadence

(number of steps per minute) and step length simultaneously under dual tasks whereas in the TD group, only the cadence was mainly affected by the dual tasks. These findings show that children with TD can be affected in one gait parameter without any other impact whereas children with CP will be affected in more aspects of their gait. A recent meta-analysis underlines the negative impact of fast-walking speed on gait parameters such as stride length and gait velocity in children with CP. Indeed, they showed greater differences as compared to a TD matched group at fast-walking speed than at self-selected pace [39]. Chakraborty et al. [39] also suggested that gait parameter variabilities and kinematic abnormalities could be a consequence of the effort provided to maintain a good stability. Finally, walking speed which has been described as functional capacity is an important parameter which can influence activities of daily living as well as the quality of life [40,41].

## Limitations

The first limit to be acknowledged was the large age span (from 8 to 16 years old). This implied a lot of differences in physical and cognitive maturation but had the advantage of representing the school-aged paediatric population as soon as their gait pattern is stabilised [42]. We reduced the induced gait and cognitive differences by matching the patients per age and sex. Also, probably because of the low GMFCS level of the participants with CP, the two groups were statistically similar in weight and height. Secondly, the biomechanical concept on which the DFE computation relies, the inverted pendulum model, needs few steps to calculate the stability. The advantage is that children had short trials. Its disadvantage is that the means were calculated based on few strides and are very sensitive to variations. This limit was constrained by the length of the gait laboratory walkway and can hardly be avoided. In order to analyse a few more strides in such settings, children would have to turn, which also implies gait stability challenges [43]. A longer trial would also increase the difficulty of the cognitive tasks, to a greater extent for the listing of fruits and animals tasks, where the repertoire is often limited. Thirdly, the cognitive score differences between the cognitive simple tasks, and the dual tasks, have to be taken into account with caution as the sitting period always occurred after the dual-task due to the need to capture the walking period first, so a learning effect may explain part of the differences.

In the future, it would be interesting to investigate the correlations between the real risk of fall and the FPE, as well as to determine the clinical minimal detectable changes which could be obtained using a mixed-methods model in different age categories. It is important to pursue research in this field to better understand the causes of the differences we observed and to be able to adapt therapies, for example by combining cognitive tasks with gait balance exercises, or to adapt the living context to the children.

## Conclusion

In this study, we showed that the gait stability of children with mild CP and children with TD is modified under dual tasks. These modifications are most likely due to the shared attention between gait and cognitive tasks. This means that cognitive and motor functions are linked, and that gait is not fully automatised. The importance of the gait compensation depends mainly upon the difficulty of the cognitive task. TD and CP children favoured a 'cognitive-first" strategy during the dual task of naming animals and fruits alternatively and increased their stability. However, children's preexisting disability, here CP, has an impact on the magnitude of their adaptations. This study underlined the impact of the child's usual activity on gait stability and encourages clinicians to consider this aspect for therapeutic management.

## Supporting information

**S1 Data.**
(XLSX)

**S2 Data.**
(XLSX)

## Acknowledgments

The authors thank the participants and their families. We also thank Nathalie Valenza for her contributions in choosing the cognitive tasks and participating in the protocol design. We thank Antoine Poncet for his help on the statistical analysis.

## Author Contributions

**Conceptualization:** Sophie Wist, Lena Carcreff, Gilles Allali, Christopher J. Newman, Joel Fluss, Stéphane Armand.

**Data curation:** Sophie Wist, Lena Carcreff, Stéphane Armand.

**Formal analysis:** Sophie Wist, Lena Carcreff, Stéphane Armand.

**Funding acquisition:** Lena Carcreff.

**Investigation:** Sophie Wist, Lena Carcreff.

**Methodology:** Sophie Wist, Lena Carcreff, Sjoerd M. Bruijn, Stéphane Armand.

**Project administration:** Sophie Wist, Lena Carcreff.

**Resources:** Stéphane Armand.

**Software:** Sjoerd M. Bruijn, Stéphane Armand.

**Supervision:** Lena Carcreff, Stéphane Armand.

**Validation:** Stéphane Armand.

**Writing – original draft:** Sophie Wist.

**Writing – review & editing:** Sophie Wist, Lena Carcreff, Sjoerd M. Bruijn, Gilles Allali, Christopher J. Newman, Joel Fluss, Stéphane Armand.

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
