## [Decision Letter · Decision Letter 0]

26 Apr 2022

PONE-D-22-05897Gait stability in ambulant children with cerebral palsy during dual tasksPLOS ONE

Dear Dr. Wist,

Thank you for submitting your manuscript to PLOS ONE. After careful consideration, we feel that it has merit but does not fully meet PLOS ONE’s publication criteria as it currently stands. Therefore, we invite you to submit a revised version of the manuscript that addresses the points raised during the review process.

While the reviewer is generally positive about the study, based on his/her and my own assessment of the ms, some adjustments and clarifications in the methods/results are necessary (see the comments). 

We look forward to receiving your revised manuscript.

Kind regards,

Yury Ivanenko

Academic Editor

PLOS ONE

Journal Requirements:

"The authors thank the participants and their families. We also thank Nathalie Valenza for her contributions in choosing the cognitive tasks and participating in the protocol design. We thank Antoine Poncet for his help on the statistical analysis. This work was supported by La Fondation Paralysie Cerebrale (Paris, France). Sjoerd M.Bruijn was funded by a VIDI grant (016.Vidi.178.014) from the Dutch Organization for Scientific Research (NWO)."

We note that you have provided funding information. However, funding information should not appear in the Acknowledgments section or other areas of your manuscript. We will only publish funding information present in the Funding Statement section of the online submission form. 

"This work was supported by La Fondation Paralysie Cerebrale (Paris, France,https://www.fondationparalysiecerebrale.org/), no grant number is available. Sjoerd M.Bruijn was funded by a VIDI grant (016.Vidi.178.014) from the Dutch Organization for Scientific Research (NWO, https://www.nwo.nl/en).

Reviewers' comments:

Reviewer's Responses to Questions

**Comments to the Author**

1. Is the manuscript technically sound, and do the data support the conclusions?

Reviewer #1: Yes

2. Has the statistical analysis been performed appropriately and rigorously? 

Reviewer #1: Yes

3. Have the authors made all data underlying the findings in their manuscript fully available?

Reviewer #1: No

4. Is the manuscript presented in an intelligible fashion and written in standard English?

Reviewer #1: Yes

5. Review Comments to the Author

Reviewer #1: Review of Manuscript ID PONE-D-22-05897 ’Gait stability in ambulant children with cerebral palsy during dual tasks’.

This is an excellent study of an important topic. I have some suggestions to improve clarity outlined below.

METHOD

• I recommend you change the description of the study design (line 13, 81). Currently it reads: ’prospective cross-sectional study with matched controls.’ These are three different designs: prospective implies that it would be a longitudinal cohort study, cross-sectional that it reflects one time point for each individual (cannot be prospective or retrospective), and finally, matched controls that it would be some sort of a case control study. In the statistical section (line 185-6) you refer to the used of the STROBE checklist for case control studies. So, I suggest you choose one of them rather than all three.

• According to the method section (line 86-9) children with CP were matched to controls based on age: ’…as well as age matched children with TD. The sample consisted of two groups, children with CP (CP group) and age-matched children with TD (TD group). Children were age matched with peers, with a range of ±1.5 years.’

However, in the discussion, limitations (line 296-7) you state they were matched by age and sex: ’We reduced the induced gait and cognitive differences by matching the patients per age and sex.’ Please revise for consistency.

• Participants. Dysarthria with difficulties coordinating the muscles used for speaking and coordination of breathing, is more frequent in children with CP. This could potentially influence the tests (cognitive tasks requiring talk: numbers, fruits, animals and be worse while walking rapidly). If none of the children had dysarthria this might be worth mentioning in the method section. If not, it could be mentioned as a limitation.

RESULTS

• Table 1. According to the characteristics of the participants, the height and especially the mean weight of the study group (CP) exceeds the values of the control group (TD) by 10 kg. This is quite unusual, and typically we see the reversed pattern with more underweight in children with CP than in TP children. Currently the weight range is reported as 21 to 107 kg in the children with CP and that is unusually high. Are these values correct? If so, please highlight also in the discussion.

• Table 3, it is interesting that children with CP seem to have the same cadence as TD children during all three conditions, even though step length is slightly shorter. I think this could be mentioned in the discussion.

• Figures. Page 6 (line 202) you refer to Figure 1A and 2B for gait stability AP and ML directions. I think this should be Figure 2A and 2B. Figure 1 illustrating Foot Placement Estimator Values is presented at page 4.

6. PLOS authors have the option to publish the peer review history of their article (what does this mean?). If published, this will include your full peer review and any attached files.

Reviewer #1: **Yes: **Elisabet Rodby-Bousquet

---

## [Author Response · Author response to Decision Letter 0]

2 Jun 2022

Dear reviewer, Dear editorial board member,

We would like to thank you for the positive feedback, and for giving us the chance to revise our work to improve its understandability and readability. We took all the comments into account and made the appropriate changes in the text with the ‘track changes’ mode. The details of the reviewer comments are available below.

Reviewer #1 

(Remarks to the Author): 

This is an excellent study of an important topic. I have some suggestions to improve clarity outlined below.

METHOD

• I recommend you change the description of the study design (line 13, 81). Currently it reads: ’prospective cross-sectional study with matched controls.’ These are three different designs: prospective implies that it would be a longitudinal cohort study, cross-sectional that it reflects one time point for each individual (cannot be prospective or retrospective), and finally, matched controls that it would be some sort of a case control study. In the statistical section (line 185-6) you refer to the used of the STROBE checklist for case control studies. So, I suggest you choose one of them rather than all three.

Thank you for this recommendation. For more clarity we adapted the text, and the study design is now expressed as “cross-sectional study”. 

• According to the method section (line 86-9) children with CP were matched to controls based on age: ’…as well as age matched children with TD. The sample consisted of two groups, children with CP (CP group) and age-matched children with TD (TD group). Children were age matched with peers, with a range of ±1.5 years.’ However, in the discussion, limitations (line 296-7) you state they were matched by age and sex: ’We reduced the induced gait and cognitive differences by matching the patients per age and sex.’ Please revise for consistency.

We would like to thank the reviewer for this comment. We revised the text, as: “The study population consisted of children with CP between the ages of 8 and 16 years with level I or II according to the Gross Motor Function Classification System (GMFCS) (23), as well as age and sex-matched children with TD. 

The sample consisted of two groups, children with CP (CP group) and age and sex-matched children with TD (TD group). Children were age matched with peers, with a range of ±1.5 years.”

• Participants. Dysarthria with difficulties coordinating the muscles used for speaking and coordination of breathing, is more frequent in children with CP. This could potentially influence the tests (cognitive tasks requiring talk: numbers, fruits, animals and be worse while walking rapidly). If none of the children had dysarthria this might be worth mentioning in the method section. If not, it could be mentioned as a limitation.

Thank you for this remark. This is a very important topic. Probably because we merely had children with low GMFCS level, we didn’t have patients with dysarthria. We added it to the method section as you suggested. “None of the children had dysarthria.” 

RESULTS

• Table 1. According to the characteristics of the participants, the height and especially the mean weight of the study group (CP) exceeds the values of the control group (TD) by 10 kg. This is quite unusual, and typically we see the reversed pattern with more underweight in children with CP than in TP children. Currently the weight range is reported as 21 to 107 kg in the children with CP and that is unusually high. Are these values correct ? If so, please highlight also in the discussion.

We thank the reviewer for this question. These values are correct. As this seems unusual, we had to have a look at the statistical differences between the groups. This was executed with a t-test. It occurs that the two groups were not statistically different. Therefore it seemed important to mention it in the results as: “The two groups had similar weight (p=0,09) and height (p=0,53).”

As suggested, we mentioned it in the discussion: “Also, probably because of the low GMFCS level of the participants with CP, the two groups were statistically similar in weight and height.”

• Table 3, it is interesting that children with CP seem to have the same cadence as TD children during all three conditions, even though step length is slightly shorter. I think this could be mentioned in the discussion.

 Thank you for this suggestion. We took this comment into great consideration and adapted the discussion as: “We expected certain differences in gait parameters between both groups. For all three tasks the step length was consistently lower in the CP group. Moreover, the difference between both groups became greater under increased dual-task difficulty. However, we observed that both groups had similar cadence, that decreased similarly in each group when the task became more difficult. It is therefore possible that in children with CP the spatial components of movement programming are more strongly impacted by dual-tasks than its temporal components, when compared to TD children.”

• Figures. Page 6 (line 202) you refer to Figure 1A and 2B for gait stability AP and ML directions. I think this should be Figure 2A and 2B. Figure 1 illustrating Foot Placement Estimator Values is presented at page 4.

We would like to thank the reviewer for this comment. We could not find this mistake, but we still adapted on the text from « Fig 2 » to « Fig 2A and 2B show the distribution of the two…” for more clarity. 

Other small changes had to be done due to new spotted mistakes or adjustments in order to meet PLOS ONE’style requirement. These are visible in the marked up copy. 

We would like to change our Financial Disclosure and Funding Information as: This work was supported by a convention (2015/1) by La Fondation Paralysie Cerebrale (Paris, France, https://www.fondationparalysiecerebrale.org/). Sjoerd M.Bruijn was funded by a VIDI grant (016.Vidi.178.014) from the Dutch Organization for Scientific Research (NOW, https://www.nwo.nl/en). The funders had no role in study design, data collection and analysis, decision to publish, or preparation of the manuscript."

We would like to adjust the amended statment as : « The authors thank the participants and their families. We also thank Nathalie Valenza for her contributions in choosing the cognitive tasks and participating in the protocol design. We thank Antoine Poncet for his help on the statistical analysis.”

We will join the data as supporting information files in the form of two excel documents. 

One of the mistakes was about the reference n°39, which is “Chakraborty S, Nandy A, Kesar TM. Gait deficits and dynamic stability in children and adolescents with cerebral palsy: a systematic review and meta-analysis. Clinical biomechanics. 2019. ». We cited wrongfully « Chakravarthy et all ».

Best regards,

Sophie Wist

---

## [Decision Letter · Decision Letter 1]

6 Jun 2022

Gait stability in ambulant children with cerebral palsy during dual tasks

PONE-D-22-05897R1

Dear Dr. Wist,

We’re pleased to inform you that your manuscript has been judged scientifically suitable for publication and will be formally accepted for publication once it meets all outstanding technical requirements.

Kind regards,

Yury Ivanenko

Academic Editor

PLOS ONE

Additional Editor Comments (optional):

Reviewers' comments:

Reviewer's Responses to Questions

**Comments to the Author**

1. If the authors have adequately addressed your comments raised in a previous round of review and you feel that this manuscript is now acceptable for publication, you may indicate that here to bypass the “Comments to the Author” section, enter your conflict of interest statement in the “Confidential to Editor” section, and submit your "Accept" recommendation.

Reviewer #1: All comments have been addressed

2. Is the manuscript technically sound, and do the data support the conclusions?

Reviewer #1: Yes

3. Has the statistical analysis been performed appropriately and rigorously? 

Reviewer #1: Yes

4. Have the authors made all data underlying the findings in their manuscript fully available?

Reviewer #1: Yes

5. Is the manuscript presented in an intelligible fashion and written in standard English?

Reviewer #1: Yes

6. Review Comments to the Author

Reviewer #1: The authors addressed all of my concerns and I endorse this manuscript for publication. This is a highly relevant topic.

7. PLOS authors have the option to publish the peer review history of their article (what does this mean?). If published, this will include your full peer review and any attached files.

Reviewer #1: No

---

## [Editor Report · Acceptance letter]

13 Jun 2022

PONE-D-22-05897R1 

Gait stability in ambulant children with cerebral palsy during dual tasks 

Dear Dr. Wist:

I'm pleased to inform you that your manuscript has been deemed suitable for publication in PLOS ONE. Congratulations! Your manuscript is now with our production department. 

Kind regards, 

on behalf of

Dr. Yury Ivanenko 

Academic Editor

PLOS ONE